# Determinants of COVID-19 Vaccine Uptake in The Netherlands: A Nationwide Registry-Based Study

**DOI:** 10.3390/vaccines11091409

**Published:** 2023-08-24

**Authors:** Joyce Pijpers, Annika van Roon, Caren van Roekel, Lisanne Labuschagne, Bente Smagge, José A. Ferreira, Hester de Melker, Susan Hahné

**Affiliations:** 1Epidemiology and Surveillance, National Institute for Public Health and the Environment, 3721 MA De Bilt, The Netherlands; 2Statistics and Modelling, National Institute for Public Health and the Environment, 3721 MA De Bilt, The Netherlands

**Keywords:** immunisation programmes, COVID-19 vaccines, socioeconomic status, migration status, political factors

## Abstract

By September 2022, the uptake of at least one dose of COVID-19 vaccine in the Dutch adult population was 84%. Ecological studies have indicated a lower uptake in certain population groups. We aimed to investigate determinants of COVID-19 vaccine uptake in the Netherlands at individual level to evaluate and optimize implementation of the vaccination program and generate hypotheses for research on drivers of, and barriers to, vaccination. A retrospective database study was performed including the entire Dutch population ≥ 18. Vaccination data (5 January 2021–18 November 2021) were at individual levels linked to sociodemographic data. Random forest analyses ranked sociodemographic determinants of COVID-19 vaccine uptake. The most important determinant was age; uptake increased until the age of 80 (67% in 18–35 years, 92% in 67–79 years, and 88% in those > 80). Personal income and socioeconomic position ranked second and third, followed by migration status. Uptake was lower among individuals in the lowest income group (69%), those receiving social benefits (56%), and individuals with two parents born abroad (59%). Our finding that age is the most important determinant for uptake likely reflects the prioritisation of elderly in the programme and the general understanding of their increased vulnerability. However, our findings also reveal important other disparities in vaccine uptake. How to best address this inequity in future vaccination campaigns requires further research.

## 1. Introduction

The COVID-19 pandemic emerged in December 2019 and has caused high rates of illness and over 6-million deaths worldwide [1]. In the Netherlands, with a population of 17.7-million people, more than 8-million infected individuals and nearly 50,000 COVID-19-related deaths have been registered up to 2022 [2]. Safe and effective vaccines were available from late 2020. Studies have indicated the beneficial impact of COVID-19 vaccination on public health in terms of the large numbers of averted deaths and hospitalisations [3,4,5]. In the Netherlands, the COVID-19 vaccination programme was implemented from 6 January 2021 onwards. Equivalent to routine childhood immunisation, COVID-19 vaccination was offered free of charge and was non-mandatory. With the aim to reduce severe disease, the vaccination programme was prioritising older age groups, people with certain medical conditions, people living in institutions, and healthcare workers [6]. By September 2022, an uptake of at least one dose of COVID-19 vaccine in the Dutch population was 84% [7].

High-risk groups that are disproportionately affected by COVID-19 include older adults and individuals with underlying medical conditions [8,9]. People living in long-term care facilities are at a higher risk of contracting and spreading the disease. Other factors such as socioeconomic status and ethnicity were also found to be associated with increased risk of COVID-19 infection and poor outcomes, underpinning the importance of homogeneous high coverage to maximise the beneficial impact of vaccination [8,10]. However, an ecological study in the Netherlands showed that the uptake of COVID-19 vaccination differed between population subgroups. The findings indicated lower vaccine uptake in neighbourhoods with a higher percentage of residents with a non-Western migration background and whose residents were more likely to vote for right-wing Christian and right-wing conservative political parties [11]. In line with those results, an ecological study investigating spatio–temporal distributions of COVID-19 vaccine uptake found lower uptake in the main urban areas and in the most religiously conservative regions of the Netherlands [12].

To date, no studies using individual-level data have been conducted to study COVID-19 vaccine uptake in the Dutch population. As individual-level data could provide more robust evidence, we aimed to investigate determinants of COVID-19 vaccine uptake in the Netherlands by linking national vaccination registration data at the individual level to sociodemographic data. Results from this study can be used to evaluate and optimize the implementation of COVID-19 vaccination programmes to increase vaccination coverage and reduce mortality and morbidity, as well as minimize health inequality. Furthermore, the results can inform health promotion activities and generate hypotheses for further research on drivers of and barriers to vaccination.

## 2. Materials and Methods

### 2.1. Study Design and Population

A retrospective database study was performed including the entire Dutch population of 18 years and older, as registered in the Personal Records Database (BRP) on 5 January 2021 (the day before the start of the COVID-19 vaccination campaign).

### 2.2. Vaccine Uptake

The outcome variable in this study was vaccine uptake, defined as having received at least one COVID-19 vaccination between 6 January 2021 and 18 November 2021 (day before start of the booster campaign). The vaccination data were retrieved from the COVID-19-vaccination Information and Monitoring System (CIMS) on 4 October 2022. CIMS is a national vaccination registration system maintained by the Dutch National Institute for Public Health and the Environment (RIVM). CIMS only includes data of people who provided informed consent to register their vaccination data. For the primary series, 93% of the vaccinees provided informed consent [13]. Vaccinated individuals who did not give informed consent could not be distinguished from unvaccinated individuals and are thus included in this study as unvaccinated. In case an individual had been infected with SARS-CoV-2 six months prior to vaccination, one dose instead of two was recommended for the primary series. However, since the CIMS database does not contain information about previous infections, completion of the primary series could not be assessed for individuals with only one primary series dose. Therefore, we chose to study vaccine uptake, defined as having received (and having provided consent for) at least one COVID-19 vaccine.

### 2.3. Determinants of Vaccination

The population database, including vaccination data, was linked to data available in the remote access environment at Statistics Netherlands (CBS). Data were linked on a person level after CBS converted citizen service numbers to a non-identifiable unique number, known as RIN (random identification number). An overview of the included determinants, and their definitions, is presented in Table 1. The majority data were most recently available for the year 2021, and some for 2020.

#### 2.3.1. Sociodemographic Determinants

Included sociodemographic determinants were age, sex, origin (captured by the two variables “country of origin” and “migration status”), socioeconomic position, personal income, household type, household car ownership, employment sector, urbanisation level, and place of residence (expressed by XY coordinates of postal code).

#### 2.3.2. Medical Risk Groups

Information on medical risk conditions was derived from Dutch national healthcare registry data. The database contained data from a claims database of outpatient specialist care utilisations and data on medication at ATC-4 code level. For selecting individuals with medical risk, we applied the method described by de Gier et al. [13]. Individuals were categorised as high, intermate, or low medical risk for severe COVID-19. The high-risk group was defined by having received care or medication for conditions associated with a high risk for severe COVID-19 [9]. People with these conditions were prioritised for the primary vaccination series. The intermediate risk group was defined by their eligibility for the annual influenza vaccination [9]. All remaining individuals were categorised as low medical risk.

In addition to underlying medical conditions, living in a nursing home and having an intellectual disability (living in residential and non-residential care) were included as separate variables in the analyses. Data on long-term care utilisation were available within a national database comprising dates and types of long-term care use at individual level. More detailed information regarding the classification of long-term care recipients is included in Appendix A.

#### 2.3.3. Voting Proportions for Political Movements

The voting proportions for political movements per neighbourhood were based on the National Elections in March 2021 in the Netherlands. We applied the classification used by Labuschagne et al. [11] that distinguished six political movements: Right-wing liberal, progressive liberal, Christian middle, right-wing Christian, progressive left wing, and right-wing conservative.

### 2.4. Statistical Analyses

Determinants were ranked according to their importance in predicting vaccine uptake at individual level using the random forest (RF) method [14]. The RF was applied to randomly selected sample of 400,000 individuals, equally divided into training and test datasets. This sample size was determined using a level at which the prediction accuracy and the ranking of the determinants stabilized, i.e., did not change visibly by taking a larger subset.

Two types of analyses were done. First, a standard RF predictor was constructed. This provided estimates of prediction accuracy, such as the PMC (probability of misclassification, i.e., the probability of predicting and individual’s status incorrectly), the sensitivity and the specificity, and the ranking of the predictor variables according to their importance, which was measured as the average increase in (worsening of) the PMC that results from the replacement of the value of a variable by a randomly chosen value. As expected, the standard RF yielded a very high sensitivity and a comparatively low specificity because there were far more vaccinated (80%) than unvaccinated individuals and the default prediction rule aims at minimizing the PMC (as opposed to the sensitivity or specificity). For this reason, a second “refined” RF, meant to yield more balanced values of sensitivity or specificity, was built. This was done by creating a standard RF with the training data and adjusting its prediction rule in the direction of the desired target based on its ROC (receiver-operating characteristics) curve, as explained in more detail in Appendix A. The performance characteristics and the variable importance of the new RF obtained by the adjusted rule were estimated by making predictions on the test data. These analyses were carried out with the R package “ranger” version 0.13.1 [15]. The results of the refined RF are presented in this paper. Since people aged 60 and above are at risk for severe outcomes of COVID-19, a second RF was performed for the population ≥ 60 years of age.

## 3. Results

On 5 January 2021, the Dutch population of 18 years and older included 14,175,650 people. The COVID-19 vaccine uptake in our dataset was 80%, with a much higher uptake in older age groups (88% in the oldest age group opposed to 67% in the youngest age group) and slightly higher among females (80% opposed to 79% in males). Lower uptake was found in individuals born abroad in the Netherlands with two parents born abroad (46%), lower income groups (69%), households with one-parent families (65%), households without a car (73% opposed to 82% with a car), individuals with low medical risk for severe COVID-19 (78% opposed to 87% in individuals with high medical risk), and individuals living in neighbourhoods with the highest voting proportions for right-wing Christian parties (40%). Uptake also varied between employment sectors (range 69–87%). A decreasing trend was observed in uptake from higher to lower education levels (range 69–85%), and from lower to higher levels of urbanisation degree (range 74–84%). Further information on uptake per variable by age group can be found in Appendix A.

### 3.1. Ranking of Determinants of Vaccination

Figure 1 shows the results of the refined RF. The performance of characteristics of the refined RF estimated from the predictions made on the test data agree with those targeted by the ROC analysis (Appendix A), which are a PMC of 30% and both a sensitivity and specificity of 70%.

#### 3.1.1. Age

The outstanding most important variable was age. The bivariate tables in Appendix A also show that uptake increases with age within the levels of every other variable. Only the highest age group (80+), for most variables, shows a slightly lower uptake than the age group 67–79.

#### 3.1.2. Personal Income and Socioeconomic Position

Personal income and socioeconomic position ranked second and third in the RF. Bivariate analyses noted that individuals with higher personal income had a higher uptake. In all percentage intervals of personal income, except for the lowest decile of 0–10%, the uptake was higher with increasing age (Appendix A). Within decile 0–10%, individuals of 67 years and older had the lowest uptake. However, this group comprised relatively few individuals. With regard to socioeconomic position, uptake was highest for people in employment and pensioners, and lowest for people with social assistance benefits. The observation that higher uptake was seen with increasing age was less clear in the group with the “another or unknown” socioeconomic position. Figure 2 shows the vaccine uptake by personal income and socioeconomic position, per age group. In people below the age of 50, uptake was higher if they had a higher personal income and worked in employment, compared to people with higher personal income in the other socioeconomic position categories.

#### 3.1.3. Population by Origin

The variable migration status ranked fourth in the RF and country of origin ranked 11th; however, in terms of an increase in PMC, their importance is comparable. Bivariate analyses (Appendix A) showed that individuals with one or two parents born in the Netherlands had a higher uptake than individuals with both parents born abroad, regardless of whether the individual was a migrant. Within the youngest age group (18–35 years), uptake was lowest for individuals born in the Netherlands with two parents born abroad, followed by migrants with two parents born abroad. Within the oldest age group, uptake is similar for all categories, except migrants with two parents born abroad for whom uptake was lower. With regard to country of origin, people from Dutch origin had the highest vaccine uptake, while people from Morocco and Middle and Eastern European countries the lowest. Figure 3 shows the vaccine uptake by origin and migration status per age group. For most countries of origin, we see a higher uptake in the higher age groups, although the differences between the age groups in some countries of origin are less clear than in others. Whether one or two parents were born abroad seems to have more influence on vaccine uptake in the age groups <50.

#### 3.1.4. Personal Income versus Country of Origin

Figure 4 shows the vaccine uptake by country of origin and personal income, per age group. Uptake was higher in higher income groups for all countries of origin. However, for some countries, this is less clear. Particularly in the 18–35 age group, there were only small differences in uptake between the personal income groups in Morocco, Turkey, the Middle and Eastern EU, and Suriname as country of origin.

#### 3.1.5. Voting Proportions for Political Movements in National Elections

Voting proportions for progressive liberal parties, right-wing liberal parties, progressive left-wing parties, and Christian middle parties ranked fourth to seventh in the RF, respectively. Figure 5 shows the vaccine uptake per age group for all political movements (the corresponding numbers can be found in Appendix A). For progressive liberal, right-wing liberal, and Christian-middle movements, the uptake increased with higher proportions of votes, particularly in individuals aged < 50. The opposite was the case for the right-wing Christian movement, with a lower uptake in people living in neighbourhoods with a higher proportion of votes. Uptake also declined slightly with higher voting proportions for right-wing conservative parties. With regard to the progressive left-wing movement, uptake was lower in people living in neighbourhoods with either the lowest or the highest category of voting proportions. For all political movements, older age groups have higher uptake as opposed to younger age groups, except for 80+.

### 3.2. Population of 60 Years and Older

On 5 January 2021, the Dutch population of 60 years and older included 4,610,000 individuals. The vaccine uptake for the 60+ population in our dataset was 90%. Compared to the general population, voting proportions for all political movements, as well as the XY coordinates of people’s place of residence, ranked higher in the variable importance for predicting COVID-19 vaccination status (Figure 6). With regard to the XY coordinates, geographically, a low uptake in the 60+ population was mainly concentrated in the most religiously conservative regions of the Netherlands (Appendix A). Although age still has predictive value on uptake in this population group, it is far less outstanding compared to the ranking of variables in the general population.

## 4. Discussion

This study presents important new insights into the determinants of COVID-19 vaccine uptake in the Netherlands. Among the adult population, the most important determinant was age. Uptake was higher in older individuals, which is consistent with previously published studies on COVID-19 vaccine uptake [7] and likely reflects the general understanding of their increased risk for severe disease. This underscores the success of prioritising elderly individuals to maximise the impact of vaccination programmes in reducing the burden of COVID-19.

After age, personal income, socioeconomic position, and origin ranked as the second-to-fourth-most important determinants of vaccine uptake. This is consistent with national registry-based studies from Norway, Sweden, and the United Kingdom, which found lower socioeconomic status and migration to be significantly associated with lower COVID-19 vaccine uptake [16,17,18,19]. Possible explanations for this are misinformation about the risks of disease due to COVID-19 and the effects of vaccination, as well as a lack of trust in authorities and the healthcare system. It is further hypothesised that factors such as language barriers, social and cultural beliefs, and religion are important [20,21]. The HELIUS study indicated that the belief that COVID-19 was exaggerated in the media was significantly associated with low vaccination intent in ethnic minority populations in Amsterdam [22]. This highlights the need for tailored communication and outreach strategies and trust [21]. Especially since it was shown that ethnic minority populations were at higher risk of SARS-CoV-2 infection and severe disease [23], further research to investigate reasons for lower vaccine uptake in those groups is important.

Voting proportions for political parties in the 2021 national elections were an important determinant for vaccine uptake as well. These findings were already reported in an ecological study on COVID-19 vaccine uptake in the Dutch population using the same data on voting proportions for political parties [11]. This study reported a positive association between vaccine uptake and higher voting proportions for right-wing liberal, progressive liberal, and Christian-middle political parties, and a negative association with higher voting proportions for right-wing Christian and right-wing conservative political parties. Voting proportions for right-wing Christian political parties have been well-described as a proxy for the Orthodox protestant religion, associated with the refusal of childhood vaccinations based on religious arguments [24,25]. Furthermore, previous studies found a positive association between higher voting proportions for right-wing conservative parties and HPV and MenACWY vaccine uptake and hypothesise these voters may have less confidence in the government, media, and social institutions [25,26].

Differences were identified in the determinants of vaccine uptake between the adult population and the 60+ group. Specifically, voting proportions for political movements ranked higher among the 60+ population, possibly reflecting generational differences in political engagement and political beliefs and values. For instance, a study by Krouwel et al. [27] revealed that individuals aged 65 and older had lower confidence in political institutions during the COVID-19 pandemic, compared to those aged 18–50. However, it is important to note that voting proportions were only available at neighbourhood level, yielding a regional effect possibly introducing ecological fallacy. Therefore, we must be cautious attributing the results directly to individual behaviour. Accordingly, we found that XY coordinates of people’s place of residence also ranked higher in the 60+ population than in the 18+ population. Further analysis at the postal code level showed that low uptake among the 60+ population was mainly concentrated in the most religiously conservative areas (i.e., regions with a relatively high proportion of orthodox protestants) of the Netherlands, while uptake was relatively high in the rest of the country. In contrast, we observed more geographical variation among the 18+ population, suggesting that other determinants played a more critical role. These findings are consistent with previously published data at the municipal level that show uptake was low among individuals ≥ 65 in a limited number of municipalities, standing in stark contrast with the high uptake in the rest of the country [28].

Interestingly, having a medical risk condition and receiving long-term care did not seem to be an important factor for vaccine uptake in our study, despite the fact that people with certain underlying medical risk conditions were prioritised in the vaccination campaign. While some studies indicated that people with a high risk for severe disease and comorbidities are more likely to accept vaccination [29], others found no association between clinical vulnerability and vaccination intention [30], or even reported more hesitancy in those groups due to the fear of possible side effects [31]. Since our data on medical risk were most recently available for 2020, recently diagnosed individuals may have been misclassified as low risk, causing an underestimation of vaccine uptake in the intermediate and high medical risk group. Furthermore, not all relevant risk conditions could be adequately identified based on the data (e.g., Down syndrome and morbid obesity). Further research investigating determinants for vaccine uptake in this specific target group is needed.

The most important strength of our study was the use of a large set of individual level data, which has significant advantage over ecological analyses on determinants of vaccination, which may be affected by ecological fallacy. Only data on political voting behaviour were not available on individual level. Furthermore, we used vaccination registration data on actual vaccine uptake rather than data on the intention to vaccinate.

This study also has limitations that should be considered when interpreting the results. First, the vaccine uptake presented in this study is an underestimation of the true uptake, as individuals who did not provide informed consent for their data to be registered were included as unvaccinated. This may have led to bias in the estimates of the effect of the determinants of vaccine uptake in different ways. In case the non-consent group has similar characteristics to the individuals who did provide consent, the true effect of the determinants is larger than the observed effect. We know older individuals provide informed consent more often than younger individuals (personal communication S. McDonald), which is similar to the positive association between age and vaccine uptake (consent group). However, information on other individual characteristics of this group is unavailable. Without this information, the extent of the bias introduced due to the informed consent cannot be investigated. Secondly, the study had limitations in terms of data quality. Education level data were incomplete for older age groups and missing for individuals that received education abroad. For medical risk and long-term care variables, the reference date was earlier in time than the reference date of the population (and the start of the vaccination campaign), causing some inaccuracies in the data. However, since the determinants we found are largely consistent with previous studies, we believe that the inaccuracies cause by data quality issues are limited. Lastly, this study could not include other potential determinants of vaccine uptake, or consent, such as religion, beliefs, and values regarding vaccination, safety concerns, language barriers, trust in the government, or distance to vaccination location, since data on these variables were not available at population level. Previous studies have shown that these factors may play a role in vaccine acceptance [32].

## 5. Conclusions

This study presents a ranking of the importance of determinants for COVID-19 vaccine uptake in the Netherlands. Our finding that age is the most important determinant for uptake likely reflects the prioritisation of older people in the programme and the general understanding of their increased vulnerability. However, our findings also reveal other important disparities in COVID-19 vaccine uptake. Lower uptake was found in individuals with lower income, lower socioeconomic status, and non-Dutch origin. How to best address this inequity in future vaccination campaigns requires further research.

## Figures and Tables

**Figure 1 vaccines-11-01409-f001:**
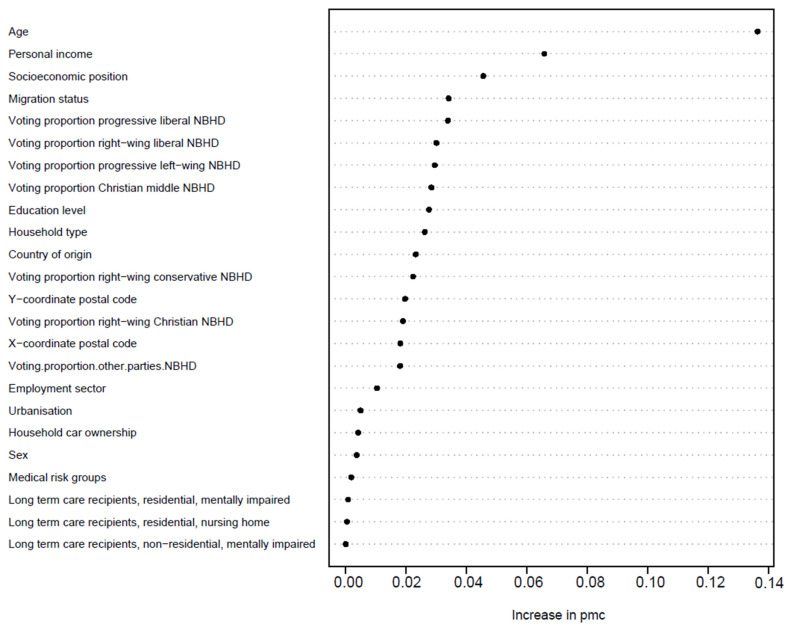
Ranking of variable importance for predicting COVID-19 vaccination status based on the refined RF. PMC = 0.30; sensitivity = 0.70; specificity = 0.70, Abbreviations: NBHD = neighbourhood.

**Figure 2 vaccines-11-01409-f002:**
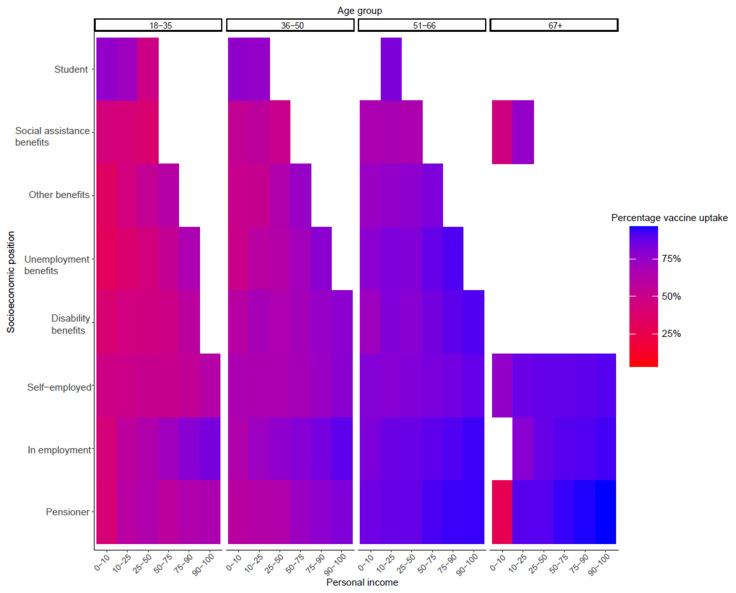
Vaccine uptake by personal income (deciles) and socioeconomic position per age group. Groups with frequencies < 100 were excluded and coloured white.

**Figure 3 vaccines-11-01409-f003:**
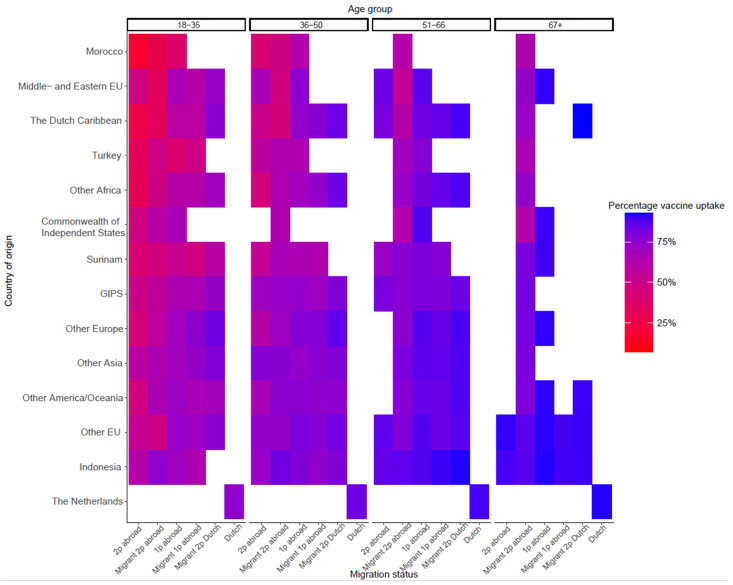
Vaccine uptake by country of origin and migration status per age group. Abbreviations: EU European Union, GIPS Greece, Italy, Portugal, Spain, 2p two parents, 1p one parent. Groups with frequencies < 100 were excluded and coloured white.

**Figure 4 vaccines-11-01409-f004:**
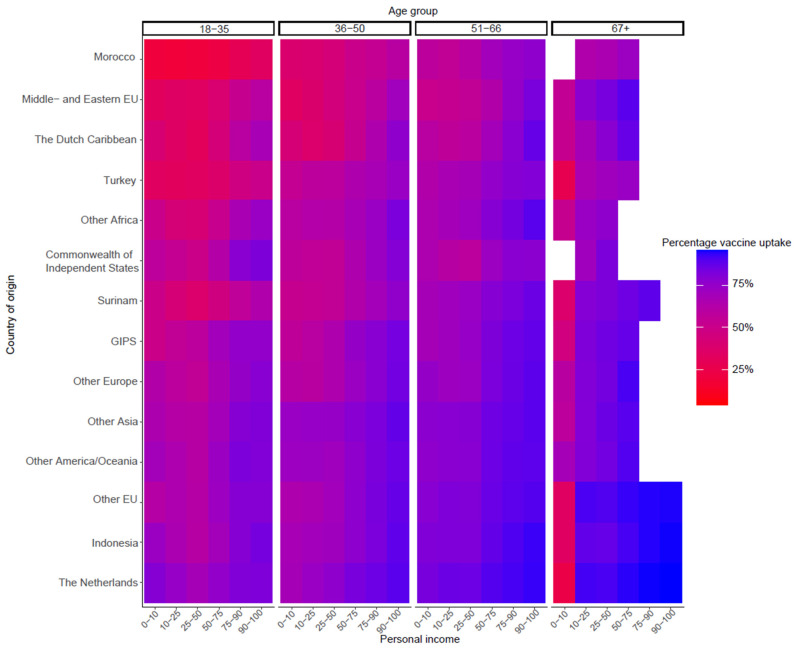
Vaccine uptake by country of origin and personal income (deciles) per age group. Abbreviations: EU European Union, GIPS Greece, Italy, Portugal, Spain. Groups with frequencies < 100 were excluded and coloured white.

**Figure 5 vaccines-11-01409-f005:**
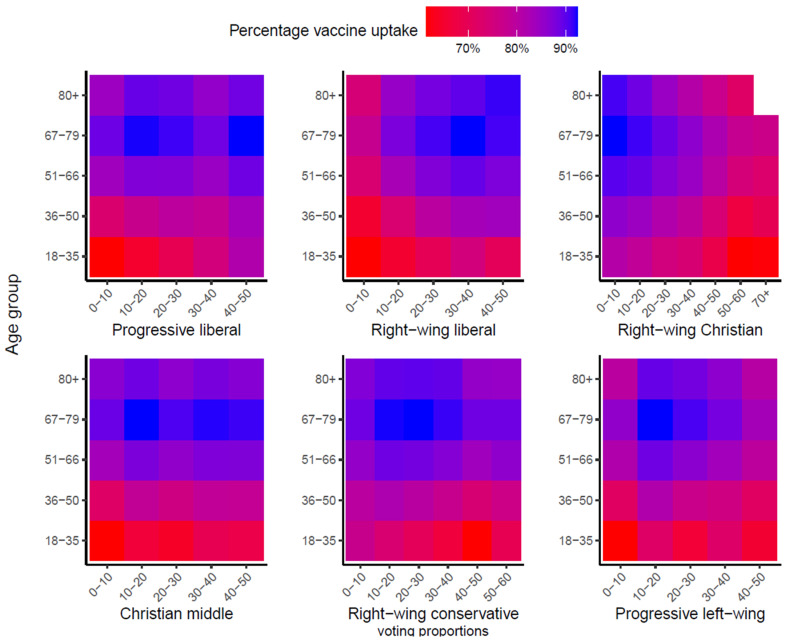
Vaccine uptake by voting proportions (%) for political movements in the 2021 National Elections, per age group. Groups with frequencies < 100 were excluded and coloured white.

**Figure 6 vaccines-11-01409-f006:**
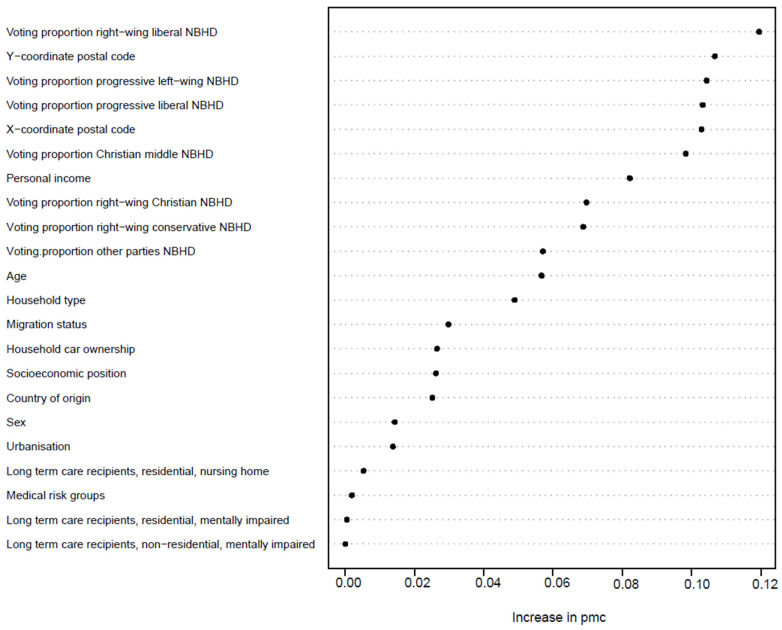
Ranking of variable importance for predicting COVID-19 vaccination status in people aged ≥60 years of age based on the refined RF. PMC = 0.30; sensitivity = 0.70; specificity = 0.70, Abbreviations: NBHD = neighbourhood, The variables education level (due to incomplete data in this age group) and employment sector were excluded from this analysis.

**Table 1 vaccines-11-01409-t001:** Characteristics of outcome variable and determinants: level of aggregation, measurement level, date of reference, and original database.

Variable	Levels	Reference Date (dd-mm-yyyy)	Database
**Individual level**			
Vaccine uptake	At least one COVID-19 vaccination registered in CIMS; 1 = yes; 0 = no ^a^	5 January 2021–18 November 2021 ^b^	CIMS
Age ^c^	Continuous	5 January 2021	CBS
Sex	0 = male; 1 = female	5 January 2021	CBS
Education level	Primary education; prevocational secondary education-basic vocational programme (VMBO-b/k), lower secondary vocational training and assistant’s training (MBO-1); prevocational secondary education—theoretical and vocational programme (VMBO-g/t), the first three years of senior general secondary education (HAVO) and pre-university secondary education (VWO); basic vocational training (MBO-2) and vocational training (MBO-3); middle management and specialist education (MBO-4); upper secondary education (HAVO/VWO); Hbo-, wo-bachelor; Hbo-, wo-master, doctor; Unknown.	5 January 2021	CBS
Country of origin ^d^	The Netherlands; Turkey; Morocco; Suriname; The Dutch Caribbean; Indonesia; Other Africa; Other Asia; Other America/Oceania; Middle and Eastern European countries within the EU; GIPS countries (Greece, Italy, Portugal, Spain); Former or associated member states of the Commonwealth of Independent States, other countries of the EU; Other European countries; Unknown.	5 January 2021	CBS
Migration status ^e^	Netherlands; Born in the Netherlands with one parent born abroad; Born in the Netherlands with two parents born abroad; Born abroad with one parent born abroad; Born abroad with two parents born abroad; Born abroad with two parents born in the Netherlands; Unknown.	2021	CBS
Socioeconomic position	In employment; Self-employed; Unemployment benefits (WW); Social assistance benefit; Other benefits; Disability benefit; Pensioner; Student; Other/Unknown.	2021	CBS
Personal income	Continuous (percentiles) ^f^	2021	CBS
Household type	One-person household; Unmarried couple without children; Married couple without children; Unmarried couple with children; Married couple with children; One-parent family; Other households; Institutional household; Other.	2021	CBS
Household car ownership	1 = yes; 0 = no	2021	CBS
Employment sector	Agriculture, Forestry and fishery; Mining and quarrying; Industry; Electricity supply; Water supply, sewerage and waste management; Construction; Wholesale and retail trade; Transportation and storage; Accommodation and food service activities; Information and communication; Financial services; Real estate activities; Professional scientific and technical activities; Administrative and support service activities; Public administration and defence; Education; Human health and social work activities; Arts, entertainment and recreation; Other service activities; Activities of household as employer, undifferentiated goods- and service producing activities of households for own use; Activities of extraterritorial organisations and bodies; Other/unemployed/unknown.	2021	CBS
Urbanisation level	Not urbanised; Hardly urbanised; Moderately urbanised; Strongly urbanised; Extremely urbanised; Unknown.	2021	CBS
X-coordinate postal code	Numeric	2021	CBS
Y-coordinate postal code	Numeric	2021	CBS
Medical risk groups ^g^	High medical risk; Intermediate medical risk; Low medical risk.	2020 ^i^	CBS
Long term care recipients, residential, nursing home	1 = yes; 0 = no	2020	CBS
Long-term care recipients, residential, mentally impaired	1 = yes; 0 = no	2020	CBS
Long-term care recipients, non-residential, mentally impaired	1 = yes; 0 = no	2020 ^h^	CBS
**Neighbourhood (NBHD) level**			
Voting proportions for political movement ^i^		2021	Open State Foundation
Right-wing liberal	Percentage		
Progressive liberal	Percentage		
Christian middle	Percentage		
Right-wing Christian	Percentage		
Progressive left-wing	Percentage		
Right-wing conservative	Percentage		
Other parties	Percentage		

Abbreviations: *CBS* Centraal Bureau voor de Statistiek (Statistics Netherlands), *CDA* Christian Democratic Appeal, *CU* Christian Union, *D66* Democrats 66, *FvD* Forum for Democracy, *GL* Green Left, *JA21* Right Answer 2021, *PvdA* Labour party, *PvdD* Party for the Animals, *PVV* Party for Freedom, *SP* Socialist Party, *Volt* Volt Netherlands, *SGP* Reformed Political Party, *VVD* People’s Party for Freedom and Democracy. ^a^ 0 = did not receive vaccination or did not provide informed consent for data to be registered in CIMS; ^b^ The extraction date was 4 October 2022; ^c^ Age was estimated as 2021, minus year of birth; ^d^ Both country of origin and migration status were included in the analysis, based on the new classification (Level 3) of population by origin by CBS. More information on the CBS 2022 classification of population by origin can be found at: New classification of population by origin www.cbs.nl/en-gb/news/2022/07/cbs-introducing-new-population-classification-by-origin (accessed on 1 August 2022); ^e^ If a person was born in the Netherlands and their mother was born abroad, country of origin was defined as the mother’s country of birth. If only the father was born abroad, their father’s country of birth was the person’s country of origin. ^f^ Percentiles are calculated by CBS Microdata based on personal income data, including the entire Dutch population in 2020; ^g^ Approximated based on healthcare utilisation and medication prescription data; ^h^ In case of rare medical conditions, data from 2016–2020 were included. ^i^ Based on information from the National Elections in March 2021 for political parties with at least two seats. Data available from the Open State Foundation URL: data.overheid.nl/community/organization/kiesraad (accessed on 15 June 2022).

## Data Availability

All data are available within CBS Microdata and can be made available under strict conditions: Microdata: Conducting your own research www.cbs.nl/en-gb/our-services/customised-services-microdata/microdata-conducting-your-own-research#:~:text=Microdata%20are%20linkable%20data%20at,strict%20conditions%20for%20statistical%20research.

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
