# Peer review of "Determinants of COVID-19 Vaccine Uptake in The Netherlands: A Nationwide Registry-Based Study"

_vaccines, 2023, doi:10.3390/vaccines11091409_

Round 1
Reviewer 1 Report
This is a well-prepared manuscript.
Some minor changes will be needed:
1/ Please add 2-3 sentences on the COVID burden in the Netherlands. This will be helpful to better understand the context of this study.
2/ Please also add 2-3 sentences on the COVID-19 vaccination program in the Netherlands (where the vaccination was available; was it free of charge? how about childhood vaccination (like MMR etc.)- voluntary or mandatory? These data will be crucial to understand the whole context.
3/ Lines 172-182 - some numbers will be helpful
4/ Figure 4 is a little bit blurred. Please improve the final version of the manuscript.
5/ Please add 2-3 sentences on the practical implications of this study (e.g., after the strengths)
Author Response
This is a well-prepared manuscript.
Some minor changes will be needed:
1/ Please add 2-3 sentences on the COVID burden in the Netherlands. This will be helpful to better understand the context of this study.
Authors: Information on the burden of COVID-19 was added to the introduction. ‘In the Netherlands, on a population of 17,7 million people, more than eight million infected individuals and nearly 50,000 COVID-19 related deaths have been registered up to 2022’. Lines 32-34.
2/ Please also add 2-3 sentences on the COVID-19 vaccination program in the Netherlands (where the vaccination was available; was it free of charge? how about childhood vaccination (like MMR etc.)- voluntary or mandatory? These data will be crucial to understand the whole context.
Authors: We added a sentence in the introduction explaining that the vaccination was offered free and non-mandatory. ‘Equivalent to routine childhood immunization, COVID-19 vaccination was offered free of charge and non-mandatory’. Lines 38-39.
3/ Lines 172-182 - some numbers will be helpful
Authors: the numbers are added to the text in lines 176-189:
‘On 05-01-2021 the Dutch population of 18 years and older included 14,175,650 persons. The COVID-19 vaccine uptake in our dataset was 80%, with a much higher uptake in older age groups (88% in the oldest age group opposed to 67% in the youngest age group) and slightly higher among females (80% opposed to 79% in males). Lower uptake was found in individuals born abroad in the Netherlands with two parents born abroad (46%), lower income groups (69%), households with one-parent families (65%), households without a car (73% opposed to 82% with a car), individuals with low medical risk for severe COVID-19 (78% opposed to 87% in individuals with high medical risk) and individuals living in neighbourhoods with the highest voting proportions for right-wing Christian parties (40%). Uptake also varied between employment sectors (range 69-87%). A decreasing trend was observed in uptake from higher to lower education levels (range 69-85%), and from lower to higher levels of urbanisation degree (range 74-84%). Further information on uptake per variable by age group can be found in Supplement 3.’
4/ Figure 4 is a little bit blurred. Please improve the final version of the manuscript.
Authors: All figures in the manuscript are adjusted to improve clarity.
5/ Please add 2-3 sentences on the practical implications of this study (e.g., after the strengths)
Authors: Our study represents a first step to understand determinants of vaccination. To be able to provide practicable recommendations to improve uptake, further research is necessary. Recommendations for further research are described in different sections in the discussion:
‘Especially since it was shown that ethnic minority populations were at higher risk of SARS-CoV-2 infection and severe disease, further research to investigate reasons for lower uptake in those groups is important.’ Lines 311-313
‘Further research investigating determinants for vaccine uptake in this specific target group is needed.’ Lines 356-358
‘How to best address this inequity in future vaccination campaigns require further research.’ Lines 392-393
Please see the attachment.

Reviewer 2 Report
Your study is straight forward and uses the information that is available in the national dataset to help understand some of the factors that result in COVID-19 vaccine uptake in the Netherlands. You accurately describe your findings as well as the limitations that are due to the retrospective nature of this type of research. While your findings may not be generalizable to all populations, your findings of lower education, socioeconomic standing and political affiliation appear to be a common theme in many Western countries and should serve as a call for targeted education for these groups for the introduction of new immunizations. I think that the issues you identify with migrant/immigrant uptake are particularly germane to 2023 as reaching these populations which are possibly distrustful of institutions and governments could serve as a major reservoir for COVID-19 and other emerging pathogens.
A few general comments:
1. Do you have any information on vaccine series completion? I know that you did not use that as your benchmark and listed why you did not report this information such as an inability to determine who had been infected or had side effects but it would be very interesting to see if the trends in determinants continued?
2. Finally, the unique blend of private/public healthcare in the Netherlands should be further discussed. You point out in your discussion that differences in access to healthcare could be on of the major factors. Do you have a mechanism to see what type of healthcare insurance/access that people have? If you are not able to report this, then, I think it would be useful to describe how healthcare and immunizations are distributed in your system so the reader can contrast that with their own.
Author Response
Your study is straight forward and uses the information that is available in the national dataset to help understand some of the factors that result in COVID-19 vaccine uptake in the Netherlands. You accurately describe your findings as well as the limitations that are due to the retrospective nature of this type of research. While your findings may not be generalizable to all populations, your findings of lower education, socioeconomic standing and political affiliation appear to be a common theme in many Western countries and should serve as a call for targeted education for these groups for the introduction of new immunizations. I think that the issues you identify with migrant/immigrant uptake are particularly germane to 2023 as reaching these populations which are possibly distrustful of institutions and governments could serve as a major reservoir for COVID-19 and other emerging pathogens.
A few general comments:
- Do you have any information on vaccine series completion? I know that you did not use that as your benchmark and listed why you did not report this information such as an inability to determine who had been infected or had side effects but it would be very interesting to see if the trends in determinants continued?
Authors: Indeed we were not able to distinguish between partly and fully vaccinated individuals. We could therefore only use the receipt of at least one COVID-19 vaccination as an outcome in the study period. We unfortunately did not have data on completion of vaccine series, and no information on infection status, that we could use. Recently, there have been some improvements in the labeling of the data. In future analyses we will probably be able to assess determinants of vaccination for different schedules. - Finally, the unique blend of private/public healthcare in the Netherlands should be further discussed. You point out in your discussion that differences in access to healthcare could be on of the major factors. Do you have a mechanism to see what type of healthcare insurance/access that people have? If you are not able to report this, then, I think it would be useful to describe how healthcare and immunizations are distributed in your system so the reader can contrast that with their own.
Authors: COVID-19 vaccination, as well as routine childhood vaccination, is free and voluntary in the Netherlands. We added a sentence to the introduction to give more context regarding this comment. ‘Equivalent to routine childhood immunization, COVID-19 vaccination was offered free of charge and non-mandatory.’ Lines 38-39.
The point regarding access to healthcare possibly has more to do with information provision/trust in authorities etc, which is also mentioned in the discussion. To be more clear, we rephrased the sentence stating access to healthcare is a possible explanation of lower vaccine uptake in migrant population groups. ‘Possible explanations for this are misinformation about the risks of disease due to COVID-19 and the effects of vaccination, and lack of trust in authorities and the healthcare system. It is further hypothesized that factors such as language barriers, social and cultural beliefs, and religion are important’. Lines 304-307
Please see the attachment.

Reviewer 3 Report
The article is interesting however there are few points tobe addressed by the authors
1. Inclusion of individuals who did not provide informed consent for their data to be included as unvaccinated may have generated an element of underestimation of the true uptake. It is not clear why these individuals were not excluded rather than included as vaccinated to avoid such possibility of bias.
2. Figures may be redesigned for more clarity
3. One of the most important missing determinant factors for vaccine uptake is religion belief
Author Response
The article is interesting however there are few points to be addressed by the authors
- Inclusion of individuals who did not provide informed consent for their data to be included as unvaccinated may have generated an element of underestimation of the true uptake. It is not clear why these individuals were not excluded rather than included as vaccinated to avoid such possibility of bias.
Authors: With the vaccination data retrieved from the national COVID-19 vaccination registration system, we could distinguish two groups: 1) individuals who received vaccination and provided informed consent for their data to be registered and 2) individuals who did not receive a vaccination + individuals who did receive a vaccination but did not provide informed consent for their data to be registered. In the second group, people with and without vaccination could not be distinguished, therefore we were unable to identify vaccinated individuals without informed consent. They were thus included among the unvaccinated. This explanation is also stated in the manuscript: ‘CIMS only includes data of people who gave informed consent to register their vaccination data. For the primary series, 93% of the vaccinees gave informed consent. Vaccinated individuals who did not give informed consent could not be distinguished from unvaccinated individuals and are thus included in this study as unvaccinated.’ Lines 79-83 - Figures may be redesigned for more clarity
Authors: All figures in the manuscript are adjusted to improve clarity.
- One of the most important missing determinant factors for vaccine uptake is religion belief
Authors: Indeed this is an important determinant for vaccine uptake. Unfortunately, we did not have access to individual data on religion/belief available at population level. From limited survey data we might have been able to retrieve some individual level information about religion, however, this comprised a very small number of individuals. Since we performed our analyses on the whole Dutch adult population, including a religion variable with missing values for most of the individuals of our study population was not possible/would have led to great bias. We added to the discussion that data on variables like religion were not available at population level. ‘Lastly, this study could not take into account other potential determinants of vaccine uptake, or consent, such as religion, beliefs and values regarding vaccination, safety concerns, language barriers, trust in the government, or distance to vaccination location, since data on these variables were not available at population level.’ Lines 381-385.
Please see the attachment.
